# A High-Performance Flexible Hydroacoustic Transducer Based on 1-3 PZT-5A/Silicone Rubber Composite

**DOI:** 10.3390/s24072081

**Published:** 2024-03-25

**Authors:** Shaohua Hao, Chao Zhong, Likun Wang, Lei Qin

**Affiliations:** 1School of Electronic Engineering, Beijing University of Posts and Telecommunications, Beijing 100876, China; wlikun1962@163.com; 2Beijing Key Laboratory for Sensors, Beijing Information Science and Technology University, Beijing 100101, China; 20192289@bistu.edu.cn (C.Z.); qinlei@bistu.edu.cn (L.Q.)

**Keywords:** piezoelectric composite, hydroacoustic transducer, flexible electrodes, cutting–filling method, underwater vehicles

## Abstract

In recent years, hydroacoustic transducers made of PZT/epoxy composites have been extensively employed in underwater detection, communication, and recognition for their high energy conversion efficiency. Despite the ease with which these transducers can be formed into complex shapes, their lack of mechanical flexibility limits their versatility across various sizes of underwater vehicles. This study introduces a novel flexible piezoelectric composite hydroacoustic transducer (FPCHT) based on a 1-3 PZT-5A/silicone rubber composite and an island–bridge flexible electrode, which can break the limitations of existing hydroacoustic transducers that do not have flexibility. The finite element method is used to optimize the structural parameters of high-performance 1-3 FPC. A large-sized (187 mm × 47 mm × 5.12 mm) FPC is fabricated using an improved cutting–filling method and packaged into the FPCHT. Compared with the planar rigid PZT/epoxy composite hydroacoustic transducer (RPCHT) of the same size, the TVR (186.5 db) of the FPCHT has increased by about 7 dB, indicating that it has better acoustic radiation performance and electroacoustic conversion efficiency. Furthermore, its electroacoustic performance exhibits excellent stability under different bending states. Therefore, the FPCHT with high electroacoustic performance is an ideal substitute for the existing RPCHT and promotes the development of hydroacoustic transducers towards flexibility and portability.

## 1. Introduction

Compared to electromagnetic and light waves, acoustic waves are considered the best carriers for long-distance information transmission in water [1]. Hydroacoustic transducers (HTs) used in hydroacoustic detection, communication, and recognition applications are commonly used sensors to achieve underwater acoustic radiation and reception [2,3,4]. The HT can be traced back to the discovery of the piezoelectric effect and inverse piezoelectric effect in quartz crystals. Subsequently, the piezoelectric material was extensively studied [5]. Up to now, piezoelectric material commonly used in hydroacoustic transducers can be divided into piezoelectric ceramics, piezoelectric crystals, and piezoelectric polymers [6]. Piezoelectric crystals have an extremely high piezoelectric coefficient d_33_ (up to 4100 pC/N) and an electromechanical coupling coefficient k_t_ (up to 0.9) [7,8]. However, their low Curie temperature and growth difficulties hinder their application in high-power and large-sized HTs [9]. Piezoelectric polymers are usually flexible, but their low d_33_ (about 20–30 pC/N) poses significant challenges in the application of emission transducers [10]. Piezoelectric ceramics are widely used in the HT due to their high piezoelectric coefficient, high Curie temperature, and stable performance [11]. Although piezoelectric ceramics are ideal materials for the piezoelectric sensitive core in the HT, they have issues such as lateral coupling and mismatch with the water [12]. To address the above issues, in 1978, R.E. Newnham et al. proposed and developed over ten connectivity patterns piezoelectric composites (PCs) [13]. Among them, the research and application of 1-3 PCs are more extensive [14]. The three-dimensional connected polymer filling between periodically arranged one-dimensional connected piezoelectric columns weakens the lateral coupling, which causes the piezoelectric columns to work in a longitudinal vibration mode. Therefore, the PCs have a higher electroacoustic energy conversion efficiency. In addition, low-density polymer fillers reduce the acoustic impedance of the PCs, which improves the sound transmission ability between the PCs and the water.

With the development of underwater vehicles towards miniaturization, the shape requirements have also been put forward for HTs. In recent years, flat, curved, cylindrical, spherical cup-shaped, and spherical piezoelectric composites and hydroacoustic transducers have been developed [15,16,17,18,19]. Unfortunately, the piezoelectric composites and transducers mentioned above are rigid and have no universality, which significantly increases the cost of underwater vehicles. Therefore, the development of flexible and conformal HT is necessary. Flexible hydroacoustic transducers require the internal piezoelectric sensitive core to have flexibility. Previous studies have confirmed that 3-3 PCs are an ideal strategy for achieving high flexibility. In 1978, D.P. Skinner et al. prepared a flexible 3-3 PZT/silicone rubber composite using the Replamineform process with a d_33_ of up to 100 pC/N [20]. Recently, Tang et al. proposed a highly flexible 3-3 PZT/PDMS composite with a d_33_ of up to 250 pm/V [21]. Although the 3-3 PCs have excellent mechanical flexibility, its low d_33_ cannot meet the high electroacoustic performance requirements of the new generation of HTs. The combination of the 1-3 PCs and flexible electrodes is another reliable strategy for achieving high mechanical flexibility. Kim et al. and Peng et al. prepared the 1-3 PZT/PDMS composite and transducer using silver nanowires as flexible electrodes, respectively [22,23]. PDMS fillers provide excellent mechanical flexibility for the PCs, and silver nanowire flexible electrodes provide stable electrical connections during bending. However, the complex preparation process and low adhesion to substrates of silver nanowires limit their application in large-scale HT [24]. Therefore, it is necessary to develop a flexible piezoelectric composite hydroacoustic transducer (FPCHT) with high electroacoustic performance and reliable flexible electrodes that can be conformal to different sizes of underwater vehicles.

Here, we develop an FPCHT with high electroacoustic performance. The 1-3 PZT-5A/silicon rubber composite and an island–bridge electrode constitute the flexible piezoelectric sensitive core of the FPCHT. Silicone rubber significantly weakens the lateral coupling between the PZT-5A and provides good mechanical flexibility for the PCs. The island–bridge flexible electrode provides a stable electrical connection of composite under arbitrary bending. The finite element method (FEM) is used to simplify the design model of the FPC and optimize the structural parameters of the FPC. The FPC was prepared by an improved “cutting–filling” method and packaged into the FPCHT. The same-size planar rigid piezoelectric composite hydroacoustic transducer (RPCHT) was prepared to verify the feasibility of the FPCHT. The performance evaluation results show that the FPCHT has a higher electromechanical conversion efficiency and sound radiation capability than the RPCHT. In addition, the FPCHT exhibits good mechanical flexibility and stable electroacoustic performance under different bending states. The FPCHT solves the defects of existing rigid hydroacoustic transducers that lack flexibility and versatility and achieves the ability to flexibly conform to different sizes or shapes of underwater vehicles. In short, the FPCHT is expected to replace the existing RPCHT and has great application prospects in the field of underwater detection, communication, and recognition.

## 2. Design of the Flexible Piezoelectric Composite Hydroacoustic Transducer

### 2.1. Structure of the Flexible Piezoelectric Composite Hydroacoustic Transducer

Figure 1a shows the structure of the flexible piezoelectric composite hydroacoustic transducer (FPCHT). The flexible piezoelectric sensitive core is packaged in a flexible polyurethane, which matches with water to provide high sound energy transmission and achieves waterproof function. The flexible piezoelectric sensitive core is composed of the 1-3 PZT-5A/silicone rubber composite and island–bridge flexible electrode (Figure 1b). The one-directional connected PZT-5A columns are arranged periodically as active materials, achieving the conversion of electroacoustic energy. Silicone rubber fillers are connected to the PZT-5A columns in three directions to form the 1-3 FPC, achieving lateral decoupling and high flexibility. The island structure of the flexible electrode is rigidly connected on both sides of the PZT-5A columns, and the U-shaped bridge is embedded in the silicone rubber to achieve the flexible connection of the island. The electrodes of the flexible sensitive core are connected to a waterproof cable, which can provide electrical energy input and electrical signal output.

### 2.2. Finite Element Analysis of the Flexible Piezoelectric Composite Hydroacoustic Transducer

As the main component of the electric–acoustic energy conversion in the flexible hydroacoustic transducer, the design of the FPC is the most noteworthy. The 1-3 FPC in the flexible piezoelectric sensitive core is formed by periodically arranging PZT-5A columns in silicone rubber. Therefore, the1-3 FPC can be regarded as a series of periodic units (Figure 2a) composed of a PZT-5A column and surrounding silicon rubber in parallel. The finite element analysis (FEA) is used to verify the above viewpoint. The models of five different numbers (1, 4, 9, 16 and 25, respectively) of the periodic units were established in ANSYS 15.0 software (ANSYS, Inc., Canonsburg, PA, USA), and corresponding admittance spectrums were obtained through harmonic response analysis on free boundary conditions (Figure 2b). The size of the periodic unit and the piezoelectric column are 7 mm × 7 mm × 5 mm and 5 mm × 5 mm × 5 mm. The PZT-5A and silicone rubber are selected for piezoelectric ceramics and flexible polymer in ANSYS, respectively, and their material parameters are shown in Table 1 and Table 2. The admittance value of the FPC at the resonance increases synchronously with the increase in periodic units, while the resonance frequency and spectrum trend remain unchanged. Therefore, the FPC can be simplified as the combination of periodic units. To verify the effect of silicon rubber on piezoelectric columns, an FPC periodic unit (7 mm × 7 mm × 5 mm) and a single PZT-5A column (5 mm × 5 mm × 5 mm) were established and analyzed in FEM. Note that the size of the PZT-5A column in the FPC periodic unit is consistent with the size of the single PZT-5A column. The admittance spectrums (Figure 2c) of the two structures show almost the same amplitude and trend, which indicates that silicone rubber has almost no electrical mechanical effect on the PZT-5A column. In addition, the vibration nephograms (Figure 2d) of the two structures also show negligible mechanical constraints of silicone rubber on the PZT-5A.

Previous studies have shown that the volume fraction and aspect ratio of piezoelectric materials are important parameters affecting the properties of the 1-3 PCs [18]. Based on the simplified model of the FPC, the periodic units of the FPC with different PZT-5A volume fractions (10–100%) were established and analyzed with ANSYS 15.0 software. Figure 3a shows the admittance spectra of the FPC at different volume fractions. Firstly, with the increase in the PZT-5A volume fraction, the vibration modes of the FPC gradually increase due to the lateral coupling effect in the PZT-5A. However, the longitudinal vibration mode (corresponding to the maximum admittance value) is the main working mode. For example, the three distinct peaks in the admittance spectrum of an FPC periodic unit (7 mm × 7 mm × 5 mm, with internal PZT-5A column (6 mm × 6 mm × 5 mm)) with a volume fraction of 70% represent the existence of three vibration modes. In the vibration nephograms (Figure 3b), the mode corresponding to 216 kHz is the longitudinal vibration mode. However, the polymer filler in the 1-3 PCs is designed to achieve lateral decoupling so that the non-longitudinal vibration is greatly weakened or even disappeared. Therefore, the high PZT-5A volume fraction in FPC is not desirable because there are more lateral couplings. Secondly, the longitudinal resonant frequency of the FPC decreases with the increase in the volume fraction. This is mainly due to the gradually apparent lateral coupling which reduces the longitudinal vibration frequency. In addition, the admittance value at the resonance is positively correlated with the volume fraction, the high admittance value indicates that the FPC has low internal loss, and the hydroacoustic transducer made of the FPC has strong acoustic radiation. Finally, Equation (1) can calculate the electromechanical coupling coefficient k_t_ for different volume fractions:(1)$$k_{t}=\sqrt{\frac{\pi }{2}\frac{f_{r}}{f_{a}}\tan\left(\frac{\pi }{2}\frac{f_{a}-f_{r}}{f_{a}}\right)}$$
where f_r_ and f_a_ are the resonant and anti-resonant frequencies in longitudinal mode, respectively. The k_t_ of the FPC is negatively correlated with the volume fraction (Figure 3c), indicating that the electroacoustic energy conversion efficiency of the FPC decreases with the increase in the volume fraction. On the other hand, the influence of aspect ratio on the FPC performance has been reported in reference [25] and exhibits similar results to volume fraction. In conclusion, the volume fraction and aspect ratio of the PZT-5A in the FPC should be moderate to meet the requirements of low frequency, high sound radiation, and high electroacoustic energy conversion efficiency of the FPCHT. Therefore, the volume fraction and the aspect ratio of PZT-5A in the FPC are determined to be 50% and 1, respectively. According to the working frequency range (<1 MHz) of the HT used in underwater detection sonar [14], the thickness of the FPC is determined to be 5 mm, and the cross-sectional area of the PZT-5A column and the periodic unit are 5 mm × 5 mm and 7 mm × 7 mm, respectively.

A large-area (187 mm × 47 mm) model of the FPC was established and analyzed in ANSYS. To verify the feasibility of the FPC, the same-size 1-3 rigid piezoelectric composite (RPC) model was analyzed and compared. The RPC is the most commonly used PZT-5A/epoxy resin composite, and their material parameters in ANSYS are shown in Table 1 and Table 2. There are some differences in the admittance spectrums of the two transducers in Figure 4a. Firstly, the resonant frequency (234 kHz) of the FPC is lower than that (246 kHz) of the RPC. Then, the FPC has a higher admittance value, which indicates that its internal losses are lower. This is because the polymer fillers in the FPC impose fewer mechanical constraints on the PZT-5A columns. Finally, the k_t_ of the FPC and the RPC are calculated by using (1) to be 0.69 and 0.5, respectively. The k_t_ of the FPC is basically consistent with the longitudinal electromechanical coupling coefficient k_33_ (0.7) of the bulk PZT, which indicates that the silicone rubber filler in FPC can better weaken the lateral coupling. Meanwhile, it also shows that the FPC has higher electromechanical conversion efficiency than the RPC. The vibration nephograms (Figure 4b) of two PCs show that FPC has better lateral decoupling and larger vibration displacement at the longitudinal resonance point compared to RPC, indicating that FPC has smaller internal losses and stronger acoustic radiation performance.

## 3. Fabrication and Measurement of the Flexible Piezoelectric Composite Hydroacoustic Transducer

### 3.1. Fabrication of the Flexible Piezoelectric Composite Hydroacoustic Transducer

The fabrication process (Figure 5) of the FPCHT can be divided into the fabrication and packaging of the flexible piezoelectric sensitive core. The flexible piezoelectric sensitive core is composed of the 1-3 FPC and island–bridge electrode, which is fabricated by a “cutting–bonding–filling” process. The detailed steps are as follows: Firstly, a high-precision slicing machine (MicroAce II, Loadpoint, Swindon UK) is used to cut a 200 mm × 50 mm × 5 mm piezoelectric ceramic block (PZT-5A, Rising Co., Ltd., Yiwu, China) into periodically arranged PZT-5A columns along the X and Y directions. The cutting depth is 2.5 mm, the cutting width is 2 mm, and the cross-sectional area of the PZT-5A column is 5 mm × 5 mm, respectively. Secondly, the strip of copper foil tape (width: 3 mm, thickness: 0.01 mm) is bonded to the surface of the PZT-5A columns along the X and Y directions. The copper foil tape (Wangxing Co., Ltd., Shenzhen, China) between the adjacent PZT-5A columns is molded using a U-shaped mold and embedded into the gap (width: 2 mm) to form a U-shaped bridge structure. Conductive silver adhesive (CD-03, Yingxun Co., Ltd., Guangzhou, China) is uniformly coated on the surface (bonding area: 5 mm × 5 mm) of PZT-5A columns and stacked copper foil tapes, and after curing, it forms a rigid island structure. Thirdly, silicone rubber (704, Kangda Co., Ltd., Qingdao, China) is filled into the gap between PZT-5A columns and cured to form a semi-finished sample after 24 h. Finally, the semi-finished sample is turned over and the above three steps are repeated to form a flexible piezoelectric sensitive core.

The flexible piezoelectric sensitive core for hydroacoustic transducer needs to be packaged to achieve waterproofing and sound energy transmission. The distributed packaging technology is used to package the flexible piezoelectric sensitive core to ensure the uniformity and consistency of the flexible waterproof layer. Firstly, the watertight cable is welded to the electrodes of the flexible piezoelectric sensitive core. Secondly, the flexible piezoelectric sensitive core is assembled with the packaging mold and perfused with liquid polyurethane. Thirdly, the semi-finished sample is formed after curing and demolding. Finally, the sample obtained in the third step is turned over and assembled with the packaging mold, and then, the third step is repeated to form the FPCHT. To further verify the feasibility of the FPC and the FPCHT, the same-size planar rigid PZT-5A/epoxy resin composite and transducer were prepared (Figure 6). The FPCHT and RPCHT operate in the thickness mode, so the thickness dimensions of each layer of the transducer are shown in Table 3.

### 3.2. Measurement of the Flexible Piezoelectric Composite Hydroacoustic Transducer

The performance parameters of the FPC are characterized by frequency, electromechanical coupling coefficient k_t_, and piezoelectric coefficient d_33_. Firstly, an impedance analyzer (4294A, Agilent, Santa Clara, CA, USA) was used to scan the admittance spectrum of the FPC in the range of 100 kHz to 400 kHz. The frequencies corresponding to the maximum and minimum admittance value in the longitudinal vibration mode are resonance and anti-resonance frequencies, respectively. Then, the k_t_ can be calculated through the resonant and anti-resonant frequency. Finally, to characterize the piezoelectric performance of the 1-3 FPC, the piezoelectric constant d_33_ was measured at a frequency of 50 Hz using a quasi-static d_33_ tester. Due to the fact that FPC is composed of two materials and has a larger size, the d_33_ values at 30 different positions on the FPC are measured. Then, Equation (2) can calculate the average d_33_ value for the FPC.
(2)$$d_{33}=\sum_{i=1}^{30}\frac{d_{33}^{i}}{30}$$
Finally, the acoustic impedance Z is an important index reflecting the transmission capacity of the acoustic energy. The mismatch between the high-impedance composite and water (1.5 MRayls) will reduce the energy transfer between the transducer and the water medium. The Z can be calculated by Equation (3):(3)Z=ρv
where ρ is the density and v is the sound velocity, which can be calculated by Equation (4):(4)$$v={2f}_{a}t$$
where f_a_ is the anti-resonant frequency and t is the thickness of the composite. In addition, the performance of the planar RPC of the same size was measured using the above method and compared with the FPC.

The electroacoustic performance parameters of the FPCHT in water are mainly characterized by the transmission voltage response (TVR), source level (SL), receiving sensitivity (RS), and directional characteristics. The underwater electroacoustic testing system (Figure 7a) is used to measure the electroacoustic performance parameters of the FPCHT. The host computer controls the signal generator (WF1946B, NF Corporation, Kanagawa, Japan) to generate a sinusoidal pulse signal. The power amplifier (M8, USA Instruments Inc., Aurora, OH, USA) amplifies the sinusoidal pulse signal and drives a sound source installed on the rotating device. The sound source converts electrical signals into acoustic signals and radiates them into the water. In the far field, a hydrophone at the same depth converts an acoustic signal into an electrical signal. The oscilloscope (MSO7034B, Agilent, USA) collects the hydrophone output and the sound source input, simultaneously, and uploads them to the host computer. After data processing, the underwater electroacoustic performance parameters of the FPCHT can be obtained. It should be noted that the measurement of acoustic radiation performance parameters (TVR and SL) and reception performance parameters (RS) is slightly different in the underwater part. When measuring the acoustic radiation performance, the FPCHT is used as the sound source, and a hydrophone (TC4035, RESON, Slangerup, Denmark) is used to receive the acoustic signal (Figure 7b). The TVR and SL values of the hydroacoustic transducer can be calculated using Equations (5) and (6).
(5)$$TVR=20\mathrm{lg}\frac{U_{0}}{U_{x}}+20\mathrm{lgd}-M_{0}$$
(6)$$SL={20\mathrm{lgU}}_{0}-M_{0}$$
where U_0_ is the output voltage of the hydrophone, Ux is the input voltage of the FPCHT, d is the distance between the sound source and the hydrophone, and M_0_ is the received sensitivity (RS) of the hydrophone. However, in the acoustic reception performance measurement system, a large-view field transducer [19] is used as the sound source, and the FPCHT and the hydrophone are placed side by side to receive the acoustic signal synchronously. The RS values of the hydroacoustic transducer can be calculated using Equation (7).
(7)$$RS=20\mathrm{lg}\frac{U_{x}}{U_{0}}+20\mathrm{lg}\frac{d_{x}}{d_{0}}+M_{0}$$
where d_0_ and d_x_ are the distances from the sound source to the hydrophone and the FPCHT, respectively. The directional characteristic of the transducer is characterized by the sound field distribution and beam angle in water. The hydrophone receives the sound signal radiated by the rotation of the FPCHT, and the sound field distribution of the FPCHT in water can be obtained.

The electroacoustic performance of the FPCHT under a flat state and three bending states is measured by the above methods. Firstly, to verify the feasibility of the FPCHT, the planar RPCHT was measured and compared with the FPCHT in the flat state. In addition, three cylindrical molds with different radii (100 mm, 150 mm, and 200 mm) were prepared to simulate the transducer carriers of different sizes. Then, the FPCHT applied to three molds was measured and compared to verify their performance stability under different bending states.

## 4. Result and Discussion

### 4.1. Performance Characterization of the Flexible Piezoelectric Composites

As the sensitive core of high-frequency hydroacoustic transducers, the performance parameters of the FPC directly affect the electroacoustic performance of the FPCHT. The complex shape of FPC indicates its good mechanical flexibility; therefore, its material properties need to be given special attention. The FEM results show that the FPC has a lower frequency and higher electromechanical coupling coefficient than the RPC. To further verify this result, the performance parameters of the FPC and RPC samples were measured and compared. Firstly, the measured admittance spectrum trends (Figure 8a) of the two composites are almost consistent with the results in the FEM. The longitudinal vibration mode is the main working mode of both composites, but the FPC has a lower resonance frequency than the RPC. Secondly, the k_t_ of the FPC is much higher than that of the RPC, indicating that the FPC has higher electroacoustic energy conversion efficiency. Finally, the d_33_ values and average value at any 30 points of the two composites are shown in Figure 8b. The high d_33_ value of the FPC means that it has higher piezoelectric properties. Therefore, the high-performance FPC is expected to be the ideal substitute for existing RPC in hydroacoustic transducers. Finally, the acoustic impedance Z of the FPC and RPC are calculated as 14.3 MRayl and 15.1 MRayl, respectively. The Z of the two composites is similar, but both are much smaller than that of the piezoelectric ceramic (about 35 MRayl), so piezoelectric composites are more compatible with water.

### 4.2. Performance Characterization of the Flexible Piezoelectric Composite Hydroacoustic Transducer

As a novel hydroacoustic transducer, the FPCHT should first focus on its underwater electroacoustic performance. To verify its electroacoustic performance, the electroacoustic performance of the FPCHT in the flat state is compared with the commonly used planar RPCHT with the same structure and size. Firstly, the FPCHT has a lower longitudinal frequency (222 kHz) than the RPCHT (246 kHz), which indicates that it has a longer sound transmission distance. The admittance spectrum trends (Figure 9a) of the two transducers are similar to those of the corresponding PCs, but their admittance values are decreased due to the large damping in water. Secondly, the k_t_ of the FPCHT and the RPCHT are 0.7 and 0.64, respectively, which means that the electroacoustic energy conversion efficiency of the FPCHT is higher. Compared to other 1-3 flexible piezoelectric composite transducers [22,23], the FPCHT in this work has a lower frequency and equivalent k_t_ (Table 4). The low-frequency sound waves generated by the FPCHT can transmit over longer distances. Meanwhile, the FPCHT also meets broadband performance requirements. Then, the acoustic radiation and reception performance of the two transducers in water were measured and compared. Figure 9b shows the TVR and the SL spectrum of the two transducers. At the resonance, the FPCHT has a higher TVR value (186.5 dB re 1 μPa/V at 1 m) than that (179.5 dB) of the RPCHT and almost the same SL value (217.5 dB) than the RPCHT. Interestingly, the input signal voltage of the FPCHT and the RPCHT at the resonance is 324 V_P-P_ and 511 V_P-P_, respectively. Therefore, the FPCHT has higher acoustic radiation performance and electroacoustic energy conversion efficiency, which benefits from lower internal losses of the FPC. There is a small difference in the RS spectrum (Figure 9c) between the two transducers, which means that the acoustic reception performance of the two transducers is the same. Finally, the sound field distribution of the two transducers at the resonance is shown in Figure 9d. Due to their large area and planar structure, both transducers exhibit sharp directionality. The beam angle (6°) of the FPCHT is 2° larger than that of the RPCHT, which may be due to the lower resonant frequency of the FPCHT. In summary, the FPCHT has a lower frequency, longer sound transmission distance, and higher electroacoustic performance than the existing RPCHT, which is more in line with the requirements of the new generation hydroacoustic transducer.

As a high-performance underwater acoustic transducer, FPCHT with high mechanical flexibility needs to be characterized by its electroacoustic performance under different bending states, as these properties reflect the stability of its electroacoustic performance when conformal with underwater vehicles of different shapes. Firstly, the admittance spectrums of the FPCHT in water under the three bending states (Figure 10a) are almost identical. The negligible difference at the resonance may be caused by the different mechanical effects of the waterproof layer on the FPC under different bending states. Therefore, the almost consistent admittance spectrum ensures that the resonant frequency and electromechanical coupling coefficient of the FPCHT remain unchanged under different TVRs and the SL spectrum (Figure 10b) in the three states is also consistent. Similar to the admittance spectrum, the two peaks in the TVR and SL spectrum indicate the existence of two vibration modes. As the radius gradually increases, the maximum TVR values at the resonance (220 kHz) are 168.3 dB, 170.4 dB, and 171.6 dB, respectively, and the maximum SL values are 209.8 dB, 212.3 dB, and 214 dB, respectively. The TVR and SL values are positively correlated with the radius, which is due to the central angle of the fan shape formed by the sound radiation surface of the FPCHT decreasing as the radius increases, resulting in more focused sound energy in the water field. The high TVR and SL values also indicate that the FPCHT maintains high acoustic radiation performance under arbitrary bending conditions, which cannot be achieved by the PVDF flexible hydroacoustic transducer [1]. In addition, the FPCHT has broadband radiation characteristics, and its working and width is about 60 kHz within −6 dB. Then, the RS spectra (Figure 10c) of the FPCHT showed consistent trends in the three states, with RS peaks around 290 kHz and 350 kHz, respectively. However, the RS value at various frequencies gradually decreases as the curvature of the FPCHT increases. This is because as the curvature of the FPCHT bending increases, the area excited by plane sound waves in thickness mode decreases, which reduces its output voltage and leads to a decrease in receiving sensitivity. In addition, the dual vibration mode coupling achieves the broadband reception characteristics of the FPCHT, with a bandwidth of up to 180 kHz within −6 dB. The FPCHT can achieve stable, high sensitivity, and broadband reception of hydroacoustic signals under different bending states. Finally, the sound field distribution of the FPCHT at the resonance (220 kHz) under three bending states is shown in Figure 10d. The beam angles within −6 dB are about 94°, 60°, and 48°, respectively. The beam angle of the FPCHT is negatively correlated with the radius, also because sound energy gradually focuses with increasing radius. The FPCHT exhibits stable sound field distribution when conformal to the hydroacoustic transducer carriers with different shapes. In summary, the FPCHT has good mechanical flexibility and exhibits high-quality and stable electroacoustic performance in different bending states, making them suitable for underwater vehicles of different sizes.

## 5. Conclusions

In this work, we propose and fabricate a flexible piezoelectric composite hydroacoustic transducer (FPCHT) with high electroacoustic performance for the first time. The flexibility of the hydroacoustic transducer allows it to effectively receive and transmit underwater acoustic signals when conformal with various sizes of underwater vehicles. The piezoelectric sensing core of the FPCHT consists of a 1-3 PZT-5A/silicone rubber composite and an island–bridge flexible electrode. The silicone rubber filler in the FPCHT contributes to excellent lateral decoupling and mechanical flexibility, thereby enhancing the acoustic radiation performance and electroacoustic conversion efficiency of the FPCHT. Moreover, the island–bridge electrode ensures a stable electrical connection even in different bending states. The FEM is used to simplify the design model of the FPC, optimize the structural parameters of the FPC, and compare and analyze the performance parameters of the large-area flexible and rigid PCs. The flexible piezoelectric sensitive core is fabricated using the “cutting–bonding–filling” method and packaged into the FPCHT. Fabrication and comparison of a planar rigid composite and transducer of the same size are performed to verify the feasibility of the FPC and the FPCHT. Consistent with the simulation results, the test results demonstrate that the FPC exhibits a higher electromechanical coupling coefficient (k_t_ = 0.7) and piezoelectric coefficient (d_33_ = 464.3 pC/N) compared to the RPC. In comparison to the RPCHT, the FPCHT operates at a lower frequency, with enhanced acoustic radiation performance (Maximum TVR is 186.5 dB) and electroacoustic conversion efficiency, and similar acoustic reception performance (Maximum RS is −186.1 dB) and directional characteristics. Furthermore, the FPCHT maintains high-quality and stable electroacoustic performance in various bending states. In conclusion, the FPCHT overcomes the limitations of existing composite transducers by offering mechanical flexibility and conformability with different underwater vehicles and serves as a promising replacement for the next generation of hydroacoustic transducers.

## Figures and Tables

**Figure 1 sensors-24-02081-f001:**
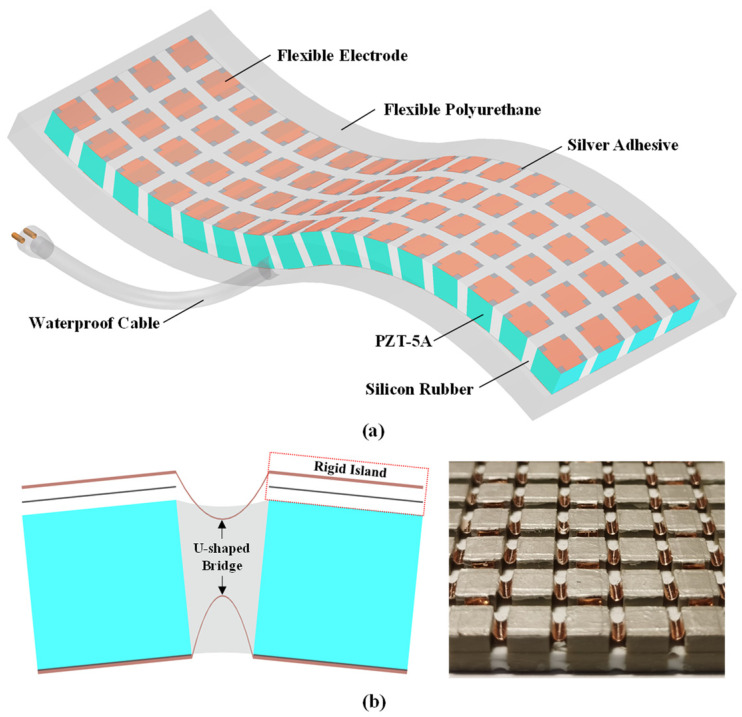
(**a**) The structure of the flexible piezoelectric composite hydroacoustic transducer. (**b**) The structure and sample of the island–bridge flexible electrode.

**Figure 2 sensors-24-02081-f002:**
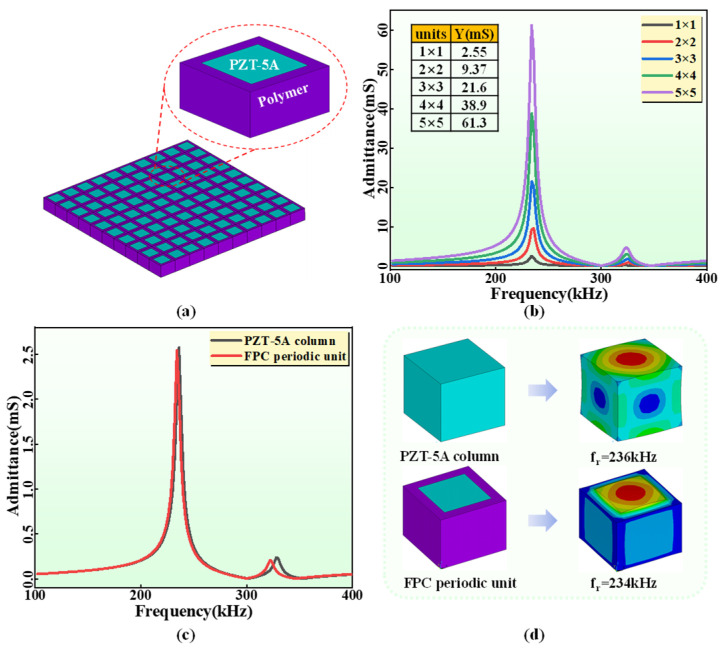
(**a**) The periodic unit of the 1-3 FPC. (**b**) The simulated admittance spectrums of different periodic units (1, 4, 9, 16 and 25, respectively). (**c**) The simulated admittance spectrums of the FPC periodic unit and the PZT-5A column. (**d**) The vibration nephograms of the FPC periodic unit and the PZT-5A column at longitudinal resonance.

**Figure 3 sensors-24-02081-f003:**
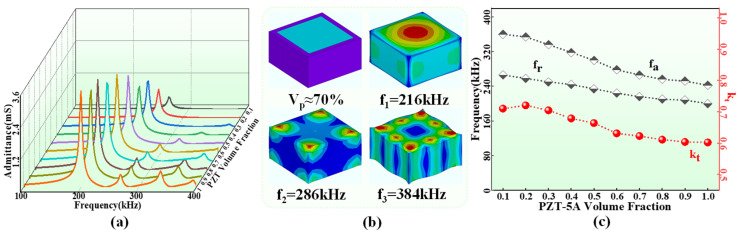
(**a**) The simulated admittance spectrums of the FPC periodic unit at different PZT-5A volume fractions (10–100%). (**b**) The vibration nephograms corresponding to the three vibration modes of the FPC periodic unit at a volume fraction of 70%. (**c**) The f_r_, f_a_, and k_t_ of the FPC periodic unit at different PZT-5A volume fractions (10–100%).

**Figure 4 sensors-24-02081-f004:**
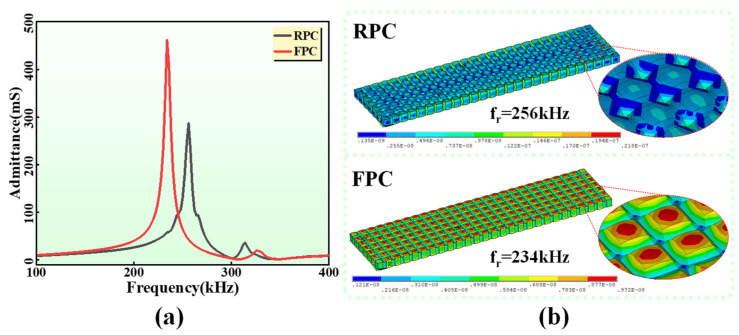
(**a**) The admittance spectrums of the large-area FPC and RPC obtained by FEM. (**b**) The longitudinal vibration nephograms of the large-area FPC and RPC.

**Figure 5 sensors-24-02081-f005:**
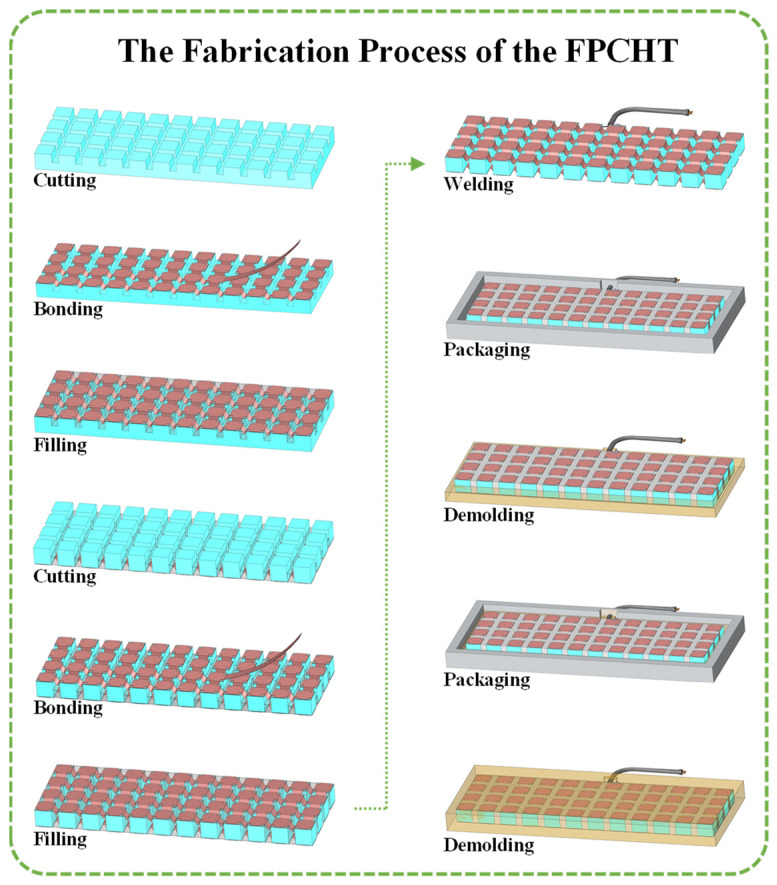
The detailed fabrication process of the FPCHT.

**Figure 6 sensors-24-02081-f006:**
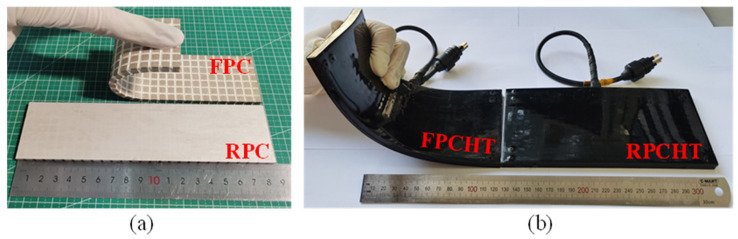
(**a**) The flexible and rigid piezoelectric sensitive core sample. (**b**) The FPCHT and RPCHT sample.

**Figure 7 sensors-24-02081-f007:**
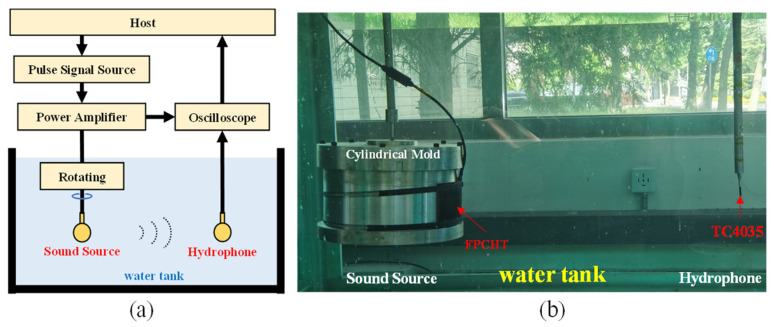
(**a**) The schematic diagram of the underwater electroacoustic testing system. (**b**) The acoustic radiation performance testing system of the FPCHT in the underwater part.

**Figure 8 sensors-24-02081-f008:**
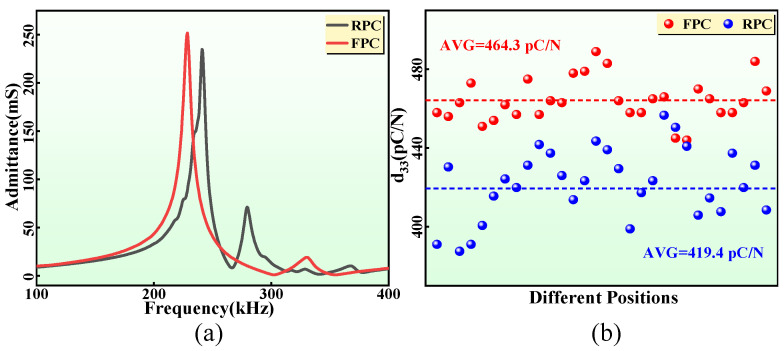
(**a**) The measured admittance spectrums of the FPC and RPC. (**b**) The d_33_ values and average value at any 30 points of the FPC and RPC.

**Figure 9 sensors-24-02081-f009:**
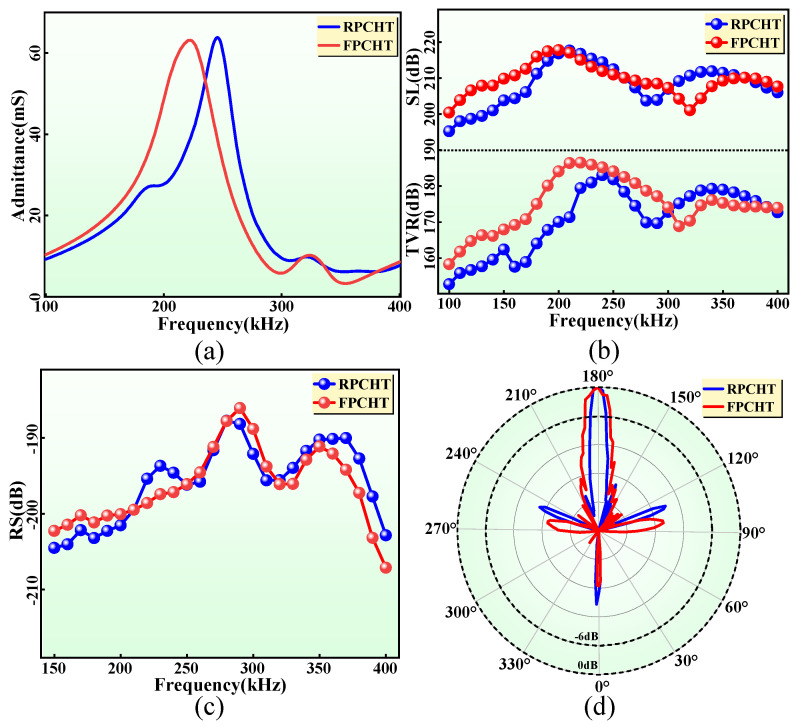
(**a**) The measured admittance spectrums of the FPCHT and RPCHT. (**b**) The TVR and SL spectrums of the FPCHT and RPCHT. (**c**) The RS spectrums of the FPCHT and RPCHT. (**d**) The sound field distribution of the FPCHT and RPCHT.

**Figure 10 sensors-24-02081-f010:**
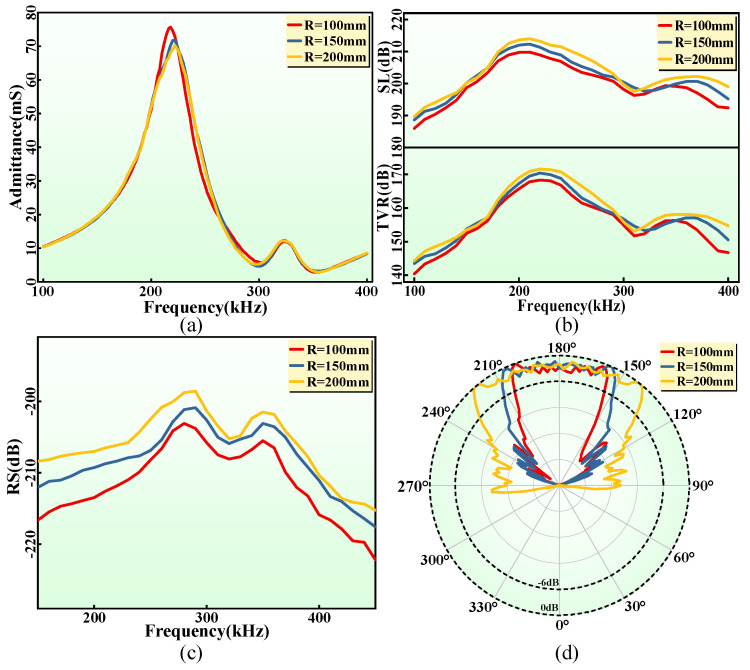
(**a**) The measured admittance spectrums of the FPCHT at the three bending states. (**b**) The TVR and SL spectrums of the FPCHT at the three bending states. (**c**) The RS spectrums of the FPCHT at the three bending states. (**d**) The sound field distribution of the FPCHT at the three bending states.

**Table 1 sensors-24-02081-t001:** Piezoelectric ceramic material parameters.

Material	Density (kg/m^3^)	Elasticity Constant (×10^10^ N/m^2^)	Piezoelectric Constant (C/m^2^)	Relative Permittivity
PZT-5A	7750	$c_{11}^{E}=12.1$ $c_{33}^{E}=11.1$ $c_{12}^{E}=7.54$ $c_{44}^{E}=2.11$ $c_{13}^{E}=7.52$ $c_{66}^{E}=2.26$	$e_{31}=-5.4$ $e_{33}=15.8$ $e_{15}=12.3$	$\varepsilon_{11}/\varepsilon_{0}=916$ $\varepsilon_{33}/\varepsilon_{0}=830$

**Table 2 sensors-24-02081-t002:** Polymer material parameters.

Polymer	Density (kg/m^3^)	Young’s Modulus (Pa)	Poisson’s Ratio
silicon rubber	1000	2.55 × 10^6^	0.48
epoxy resin	1050	3.6 × 10^9^	0.35

**Table 3 sensors-24-02081-t003:** The thickness of each layer in the FPCHT and RPCHT.

Thickness	FPCHT	RPCHT
Transducer	9.56 mm	9.5 mm
Sensitive core	5.12 mm	5.04 mm
Island-bridge electrode	0.06 mm	0.02 mm
Waterproof layer	2.22 mm	2.23 mm

**Table 4 sensors-24-02081-t004:** Performance comparison of flexible transducers.

Flexible Transducers	Frequency (kHz)	Bandwidth (−6 dB)	k_t_
PZT-5H/PDMS [22]	1700	49%	0.74
PZT-5A/PDMS [23]	4830	46.9%	0.7
PZT-5A/Silicone rubber (this work)	222	31.5%	0.7

## Data Availability

No new data were created.

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
