# Peer review of "A High-Performance Flexible Hydroacoustic Transducer Based on 1-3 PZT-5A/Silicone Rubber Composite"

_sensors, 2024, doi:10.3390/s24072081_

Round 1

Reviewer 1 Report

Comments and Suggestions for Authors

This work is devoted to research piezoelectric composite hydroacoustic transducers have been extensively employed in underwater detection, communication, and recognition for their high energy conversion efficiency in recent years. One of problems to use these transducers their lack of mechanical flexibility limits their versatility across various sizes of underwater vehicles. In this paper authors a flexible piezoelectric composite hydroacoustic transducer (FPCHT) with high electroacoustic performance has been fabricated using the 1-3 PZT-5A/silicone rubber composite serves as the flexible sensitive core. Using the finite element method authors  to optimize the structural parameters  of high-performance 1-3 FPC. Using  an  improved cutting-filling method was fabricated a large-sized  FPC  and packaged into the FPCHT. Compared with the planar rigid piezoelectric composite hydroacoustic transducer (RPCHT) of the same size, the FPCHT was displayed excellent acoustic radiation performance and electroacoustic conversion efficiency.

The introduction of the paper provide sufficient background and include all relevant references, all the cited references are relevant to the research, the research methods are adequately described, the obtained results are clearly presented and I recommend this paper to publication without any corrections.

Author Response

Thank you for your comment.

Reviewer 2 Report

Comments and Suggestions for Authors

The article "A High-Performance Flexible Hydroacoustic Transducer Based" shows new results obtained for flexible hydroacoustic transducer. The paper is well written, contains theoretical and measuring parts. All results are described. Nevertheless there are some remarks:

  1.     The following statement «However, their low Curie temperature and growth difficulties hinder the application in high-power and large-sized HT» contradicts previously published data. Some single crystalline piezoelectrics, for example, langasite, reaches 500°C (for instance, 10.1109/ULTSYM.2010.5935533 for langasite). Such materials also could be grown using well-known Czochralski method.
  2.      How possible is a short circuit between the island-bridge electrodes?
  3.     Fig.2 shows results obtained for periodic units of FPC. Nevertheless, dimensions of unit cells are unknown. What is the influence of a cell dimensions on simulated parameters?
  4.  There some typos and contradictions. E.g. it looks like the sentences in lines 143-151 contradict to each other. Sentences in lines 350-354 are the same. Please, check the text.     
  5.  The authors provide no formula explaining obtained results in Section 4.

Reviewer 3 Report

Comments and Suggestions for Authors

Content can be found in the attachment

Comments on the Quality of English Language

There are suggestions related to English writing mentioned in the attachment above.
